# *WDR36*-Associated Neurodegeneration: A Case Report Highlights Possible Mechanisms of Normal Tension Glaucoma

**DOI:** 10.3390/genes12101624

**Published:** 2021-10-15

**Authors:** Elana Meer, Tomas S. Aleman, Ahmara G. Ross

**Affiliations:** 1Department of Ophthalmology, University of Pennsylvania, Philadelphia, PA 19104, USA; Elana.Meer@pennmedicine.upenn.edu (E.M.); aleman@pennmedicine.upenn.edu (T.S.A.); 2FM Kirby Center for Molecular Ophthalmology, Scheie Eye Institute, University of Pennsylvania Perelman School of Medicine, Philadelphia, PA 19104, USA

**Keywords:** low tension glaucoma, *WDR36*, ganglion cell layer

## Abstract

*WDR36* is one of a number of genes implicated in the pathogenesis of adult-onset primary open angle glaucoma (POAG). Here we describe in detail the phenotype of a patient with pathogenic variation in *WDR36* who presented with a protracted history of central vision loss. On exam visual acuities were at 20/100 level, had a tritan color defect and showed central arcuate visual field defects on visual field testing. Enlarged cup-to-disk ratios with normal intraocular pressures were associated with severe thinning of the ganglion cell layer (GCL) and retinal nerve fiber layer consistent with a clinical diagnosis of normal tension glaucoma. Full-field electroretinograms revealed a severe inner retinal dysfunction with reduced amplitudes and remarkably delayed timings of the b-wave, but preserved photoreceptor (a-wave) function. The pattern described herein recapitulates some of the findings of an animal model of *WDR36*-associated POAG and suggests a mechanism of disease that involves a retina-wide inner retinal dysfunction and neurodegeneration beyond the GCL. Further detailed structural and functional characterizations of patients with a pathogenic variant in the *WDR36* gene are required to confirm these findings.

## 1. Introduction

Glaucoma, a chronic progressive optic neuropathy, currently affects 70 million people worldwide with 10% of affected individuals rendered irreversibly blind [1]. This is the most common optic neuropathy and is the leading cause of blindness worldwide [2]. Clinically this is a heterogenous group of disorders that is diagnosed based on characteristic changes in the optic nerve head that correspond to functional changes, often reflected as “blind spots” on visual field analysis [3]. While standard of care managing this debilitating neurodegenerative disease is lowering the intraocular pressure (IOP), this does not address the root cause of the disease, which is optic nerve degeneration. Treatment of the disease is currently limited to methods that reduce intraocular pressure (IOP), such as arduous topical medications, laser trabeculoplasty, and surgery [4,5]. Despite effective IOP lowering treatment, the disease progresses in a significant number of patients [6]. This suggests a need for a better understanding of the molecular mechanisms that contribute to the glaucoma phenotype to develop more precise and targeted medicine for better treatment.

Genetics is considered to play an influential role in understanding the fundamental causes of this very common neurodegenerative disease. Many of the more previously and extensively studied genes such as myocilin, optineurin, and *CYP1B1* have been important clues to address the etiology of glaucoma. Although, over the past 10 years, it is recognized that these variations account for less than 10% of patients presenting with disease [7]. This has led to numerous large scale genetic studies over the past 5 years, that involve analysis of a phenotype, blood or saliva testing, family history, and genome-wide association studies in patient groups that has aided in the discovery of more common genetic risk factors [8,9]. While this moves the pendulum closer to understanding the pathophysiology of the disease, certain groups still need to be represented in the analysis to direct our understanding of the disease and move the aim to more personalized treatment [10,11].

## 2. Case Report

A 70-year-old man with a medical history significant for Parkinsonism, who was a non-smoker, minimal drinker with a 10-year history of insidious, progressive, bilateral painless central vision loss. His symptoms began with a “fog” over his vision. He was an avid golfer and described the onset of his symptoms as “being unable to track a golf ball to its final location” after hitting a line drive. His symptoms were “bothersome” but stable, until a year prior to presentation to us, he began to experience difficultly seeing at night, particularly as it related to driving. He also stopped golfing due to progressive visual impairment. His symptoms prompted neuro-ophthalmic evaluation resulting in a diagnosis of an optic neuropathy due to the appearance of his nerves. He had an unremarkable brain and orbital MRIs. He also had a normal comprehensive retinal examination with cupped optic nerves in both eyes but that was otherwise unremarkable. He did not note a family history of consanguinity, however had an extensive family history of profound vision loss in two maternal aunts and three cousins with severe blindness striking as early as 55 years of age. At the time of his initial examination at age 61, his best corrected visual acuity was 20/20 in both eyes. IOPs were in low teens in both eyes. Gonioscopic examination revealed an anterior angle open to scleral spur in both eyes. There was no afferent pupillary defect and color vision in both eyes was normal. His optic nerves were reported as cupped and pale. He had a normal brain MRI and testing for nutritional or toxic optic neuropathies and genetic testing for optic neuropathies (Table 1). A comprehensive expanded retinal panel that used next generation sequencing to screen for mutations in >800 genes (panel J905, GeneDx, Gaithersburgh, MD, USA) was used. Of note, genetic testing reported no pathogenic mutations in *LHON* or *OPA1*.

Due to a progressive and insidious central vision loss he sought a second opinion with us at 69 years of age. On examination his best corrected visual acuity was 20/125 in the right eye and 20/50 in the left eye. Fundus exam revealed cupped and pale nerves in both eyes (Figure 1).

There were few small depigmented round lesions in the parafovea of both eyes, but otherwise an unremarkable retinal exam, including normal vasculature and retinal periphery. Spectral domain optical coherence tomography (SD-OCT) revealed severe thinning of the retinal nerve fiber layer (RNFL) in the superior and inferior sectors (greatest inferiorly) of the nerve corresponding to a thinned neuro-retinal rim along the same regions and a large (~0.7) cup-to-disk ratios (Figure 2A). The RNFL thinning in the peripapillary retina may be tracked as arcuate defects toward a very thin pericentral GCL (Figure 2B).

Detailed phenotyping revealed a deutan−tritan defect on Farnsworth D15 color vision testing (Figure 3A). Goldman kinetic visual fields were normal in peripheral extent with a minor superior depression while fundus perimetry demonstrated an unsteady central fixation associated with reduced foveal sensitivities as well as and deep arcuate defects in nasal pericentral retina most dense inferiorly which co-localized with the areas of GCL and RNFL thinning that tracked to the peripapillary retina (Figure 3A). Electrophysiologic testing was pursued to try to find an explanation for the central vision loss. The b-wave of the full-field electroretinogram (ffERG), which reflects the function of the inner retina, particularly bipolar cell function, were markedly reduced in amplitudes and delayed in timing for rod- and cone-mediated recordings (Figure 3B). These inner retinal abnormalities in the light-adapted ERG conferred a particularly abnormal configuration compared to the normal waveforms. By contrast, the photoreceptor-mediated a-wave was perfectly normal for cone- and dark-adapted responses (Figure 3C). Multifocal electroretinograms (mERG) confirmed this pattern of predominantly inner retinal dysfunction. The abnormalities led to a comprehensive genetic screening including of 856 genes, which revealed a c.1064 A > C pN355S pathogenic variant in *WDR36*. A central SD-OCT was performed to explore the structural correlate to these retina-wide functional abnormalities (Figure 3C). A location 2 mm in the nasal retina was chosen. Measurements were obtained at 2 mm from the foveal center where there was only minimal lateral displacement of the inner retina in relationship to the distal photoreceptors from which they received their input. Longitudinal reflectivity profiles were used to ensure proper segmentation of the main nuclear layers. Significant GCL thinning and a borderline thin INL with otherwise normal photoreceptor ONL was documented, consistent with the mainly inner retinal abnormalities detected by electroretinography (Figure 3C). The findings partially recapitulatde the findings in an animal model of *WDR*-associated retinal degeneration [12].

## 3. Discussion

In this study, we present a case report of patient with progressive, slow and insidious vision loss found to have a pathogenic variant of the *WDR36* gene identified by a comprehensive expanded retinal panel from GeneDx. Eye exam was notable revealed visual acuity of 20/20 bilaterally, low-normal intra-ocular pressures, cupping, and optic pallor bilaterally at time of first visit, with a normal brain MRI. However, over 10 years, the patient’s vision loss progressed 20/150 in the right eye and 20/50 in the left with arcuate defects on visual fields, vertical thinning on OCT, consistent with profound retinal ganglion cell and retinal nerve fiber layer loss bilaterally. There were remarkable electrophysiologic abnormalities pointing to an obvious inner retinal, post-receptoral dysfunction. Comprehensive genetic testing revealed a likely pathogenic variant in *WDR36*. Segregation of the phenotype in the patient’s family could not be confirmed as family members were not accessible to testing at this point and the patient was subsequently lost to follow up. Pathogenic variants in *WDR36* gene in a mouse model caused severe synaptic changes within the inner retina leading to a progressive retinal degeneration, reminiscent to what we describe in our patient [12].

Primary open-angle glaucoma (POAG) is a complex disease resulting in a characteristic degeneration of the optic nerve through retinal ganglion cell death. Its phenotype results in characteristic damage to axons in an arcuate pattern that results in functional changes in visual fields. While elevation in intraocular pressure is not a prerequisite for the diagnosis of glaucoma, but is the primary risk factor for the progression of the disease [13]. Aside from phenotype, glaucoma has been associated with multiple susceptibility genes and environmental factors, including WDR36 [14,15,16,17,18]. WDR36 encodes a protein of unknown function, a member of the WD repeat protein family involved in cell cycle progression, signal transduction, apoptosis, and gene regulation [14,15,19,20]. More recently, WDR36 has been described as a causative gene for adult onset POAG [21]. Located on the cytogenic band, at the 5q22.1 location, WDR36 is coregulated with IL2 involving T cell activation, and is highly expressed as 5.9 and 2.5 kb transcripts in ocular tissue such as the lens, iris, sclera, ciliary muscles, ciliary body, trabecular meshwork, retina, optic nerve) [15,19,21,22,23,24,25]. While the exact function is still debated, depletion of WDR36 mRNA in cultured cells causes apoptotic cell death with decreased 21S rRNA and delay of 18S rRNA maturation [19,20]. WDR36 knockdown in zebrafish has demonstrated the gene’s importance in nucleolar processing of 18srRNA required for ribosome biogenesis [20], as well as in the p53 stress response pathway with a lack of WDR36 leading to disrupted nucleolar function [14]. This suggests the importance of this gene in cell survival and function not just limited to the eye.

In humans, the WDR36 gene has shown varying levels of correlation with POAG diagnosis and severity. Monemi et al. (2005) found a locus for POAG on 5q with 4 variants in the WDR36 gene among 17 unrelated POAG patients, 11/17 with high pressure and 6/17 with low-pressure glaucoma (variants or mutations absent in 200 normal control chromosomes) with residues conserved between WDR 36 orthologs in mouse, rat, dog chimp and human [21]. Monemi et al.’s results demonstrated WDR36 gene expression in the lens, iris, ciliary muslces, ciliary body, trabecular meshwork, retina and optic nerve established by RT-PCR with four pathogenic variants in the 5q22.1 GLCIG gene (N355S, A449T, R529Q and D658G) causative for adult-onset POAG with implications for both high- and low-pressure glaucoma [21]. Fingert et al. (2007) did not show an association between variations in the WDR36 gene and POAG in two large cohorts of patients with POAG and ethically matched controls in the Iowa college of medicine database [26], while another investigation published by Footz et al. (2009) suggested that WDR36 sequence variate can lead to altered phenotype in polygenic forms of glaucoma [27]. There is some contradictory evidence in previously published reports on the effect of WDR36 gene mutations, and its allelic variants on the development of POAG. Hewitt et al. (2006) found WDR36 D658G to be a neutral variant in the Australian population [28]. Weisschuh et al. (2007) found that WDR36 gene variants are only rare causes of NTG in the German population [29] corroborated with a study by Pasutto et al. suggesting it may be only a minor contributing variant in this same population [25,29]. Hauser et al. (2006) found that abnormalities in the WDR36 were not sufficient to cause POAG but can contribute and be a glaucoma modifier gene associated with greater severity of illness [18]. As expected, when associating a common phenotype, with a genotype, its prevalence and importance becomes evident based on the study population.

Studies have continued to demonstrate equivocal effects with some showing a lack of clear effect in particular populations [24,30,31,32] while others have demonstrated the WDR36 gene to be a contributing risk factor for disease progression and severity [24,27,33,34,35,36,37,38]. There is a known genetic interaction between WDR36 and p53 variants in POAG susceptibility [14,39] and a clearly important functional role in retina homeostasis [12]. Therefore, given WDR36′s potential as a causative gene for adult-onset POAG in some populations and a modulator/coregulated driver in ocular disease, it is essential to further understand the phenotype of expression for pathogenic variants in the WDR36.

## 4. Conclusions

Although there is some conflict in the literature regarding the role of *WDR36* variants and the genetic contribution to the glaucoma phenotype we present a case of a patient with a clear glaucomatous optic neuropathy confirmed by GCL losses. Interestingly, our patient showed obvious inner retinal functional abnormalities that may be similar to the abnormalities reported in a murine model of the WDR36-associated disease. A search for a similar structural and functional phenotype in other patients as well as the segregation analysis of the phenotype in families with similar molecular defects are needed to confirm the pathogenicity of *WDR36* variants in similar situations. Genetic studies will prove helpful in unmasking critical molecular mechanisms that will add in staging, predicting progression, and developing personalized therapies for this debilitating disease. While advancing to this specialized and potentially beneficial areas of treatment, the prevalence of a genetic variants studied for treatment is both population and phenotype dependent and should be considered when developing genetic studies.

## Figures and Tables

**Figure 1 genes-12-01624-f001:**
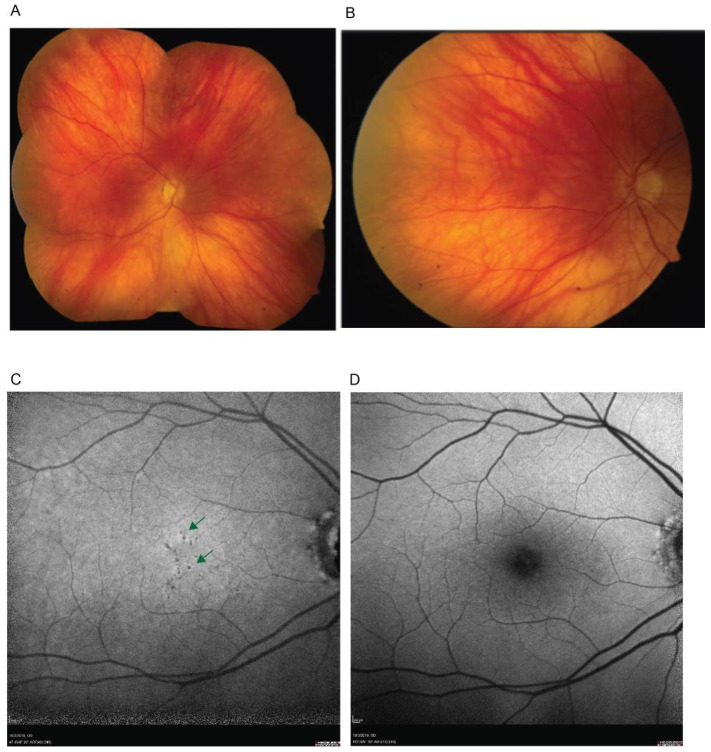
En face retinal imaging demonstrating retinal findings. (**A**). A color fundus photography photomontage shows a normal blonde fundus. (**B**) A magnified image of the central retina demonstrates small faintly depigmented round lesions (green arrows) near the foveal center. (**C**,**D**). Fundus autofluorescence imaging using near-infrared (NIR-FAF) (**C**) and short-wavelength (SW-FAF) (**D**) excitation lights. Arrows on the NIR-FAF image (**C**) points to round dark, hypo-autofluorescent, depigmented lesions; SW-FAF image (**D**) is within normal limits. Only right eye shown for clarity, left eye had similar findings.

**Figure 2 genes-12-01624-f002:**
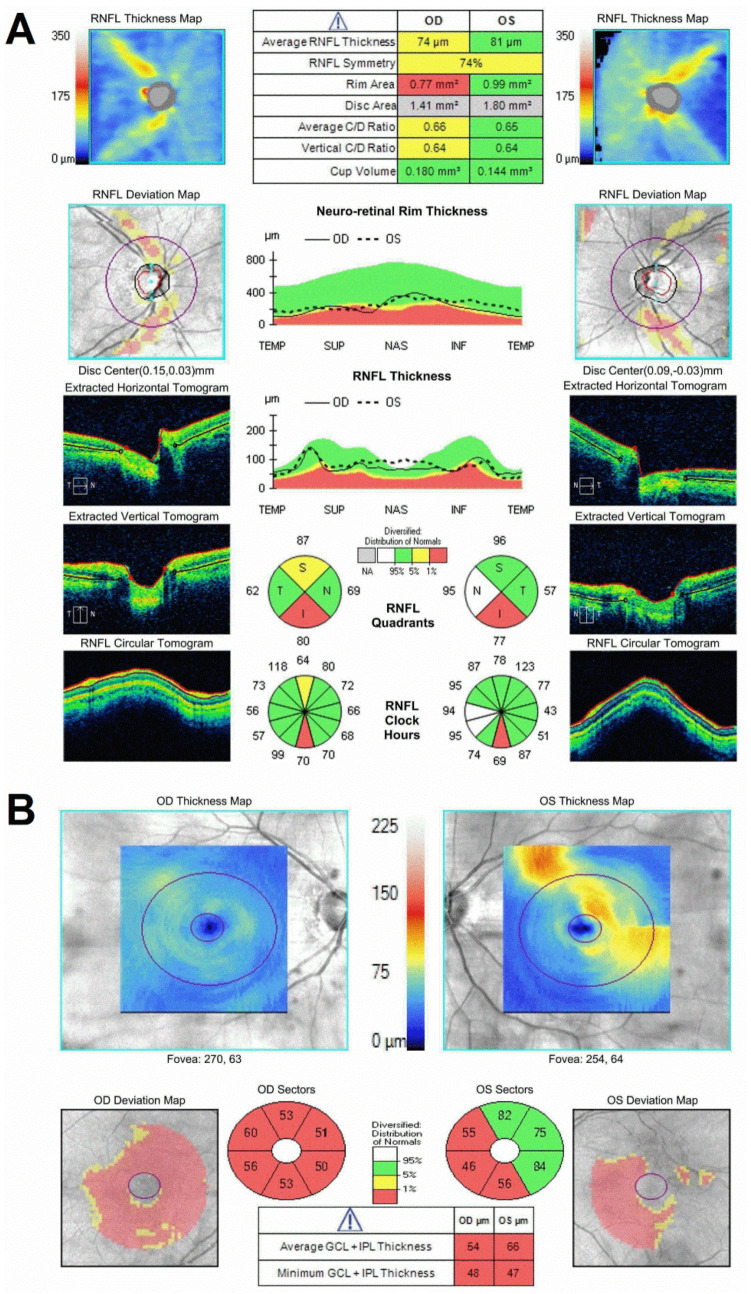
Optic nerve and retinal photography by OCT demonstrating vertical optic nerve atrophy with associated thinning in the retinal ganglion cell layer. SD−OCT of the peripapillary and central retina in the *WDR36*-positive patient. (**A**) SD−OCT of the peripapillary retina in the patient. Sides (left and right columns) are plots of the topography of the peripapillary RNFL thickness as raw thickness values (top row) and as deviation maps (second row) compared to a normative database for the right (left panels) and left (right panels) eyes of the patient. The bottom three rows of panels below are horizontal (top) and vertical (middle) cross-sections SD−OCT scans through the optic nerve, as well as circular scans cross-sectional tomograms around the optic nerve. The middle column is summary parameters. (**B**) SD−OCT of the central and pericentral retina in the patient. Top panels are topographic maps of the GCL thickness (there are segmentation artifacts shown as localized thickened GCL in the left eye). Bottom panels are GCL thickness summary parameters compared to a normal distribution database.

**Figure 3 genes-12-01624-f003:**
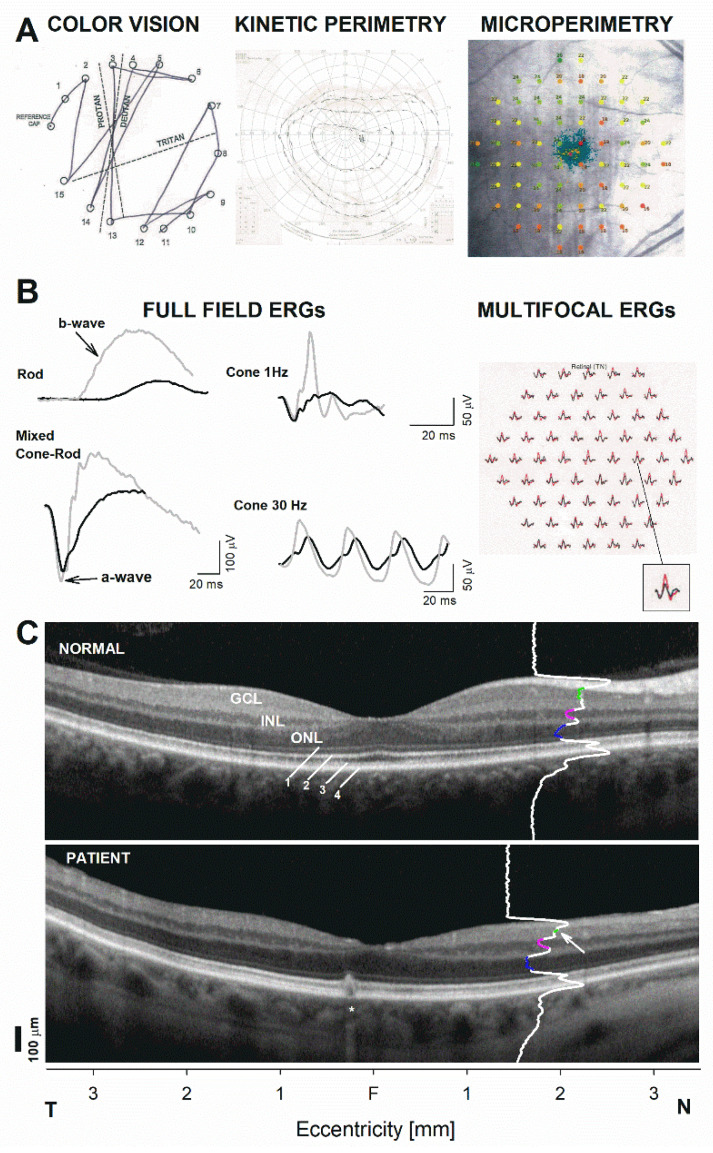
Detailed functional and structural phenotyping of the *WDR36*-positive patient using color vision and perimetric functional analysis, followed by electrophysiology. Only one eye is shown for clarity; results are nearly identical in the contralateral eye. (**A**) Left panel: color vision documented with a Farnsworth–Munsell D15 test; middle panel: Goldman kinetic perimetry to three stimulus conditions (V-4e, III-4e and I-4e); right panel: dark-adapted (>30 min), fully dilated fundus perimetry or microperimetry (MAIA, CenterVue, Padova, Italy) from our patient. Sensitivities are shown as numerical values next to the testing (a 68 point, conventional ‘10-2 protocol’ grid), as well as color-coded sensitivity losses compared to a normative database (green symbols, ≥25 dB, considered a rough lower limit of normal across the grid). Blue-green traces near the center portray prefer locus and fixation excursions. (**B**) Standard electroretinography elicited using a commercially available system (Espion e3, Diagnosys LLC, Lowell, MA, USA). Standard full-field electroretinograms (ffERGs) recorded in scotopic (left panel) and photopic (middle) conditions, as well as multifocal electroretinograms (mERG) (right) across the central 20 degrees are shown. ff-ERGs from the patient overlap representative normal responses, gray traces. m-ERG traces in the patient, black traces, are also compared to average normal traces (red). Mixed cone-rod response is elicited with a standard flash (nominal 3.0 setting of standard). A magnified comparison from a parafoveal location is shown to the bottom right to illustrate a perfect match of the photoreceptor a-wave component which contrasts with the inner retina, b-wave, component. (**C**) Horizontal SD-OCT (Spectralis, Heidelberg Engineering, Heidelberg, Germany) through the foveal center in the patient compared to a representative normal subject. Nuclear layers are labeled: outer nuclear layer = ONL, inner nuclear layer = INL, ganglion cell layer = GCL. Outer retinal sublaminae are labeled as: 1. Outer limiting membrane (OLM), 2. Inner segment ellipsoid band (EZ), 3. Interdigitation of the photoreceptor outer segment tip with the apical RPE (IZ), 4. Retinal pigment epithelium and Bruch’s membrane (RPE/BrM). A longitudinal reflectivity profile is shown overlapping the cross section, 2 mm in nasal retina. Colored segments denote the extent of the signal trough that corresponds to the ONL (blue), the INL (pink), and the GCL (green). Arrow points to a thinned GCL, asterisk denotes incidental pigment epithelial detachment in the temporal juxtafovea. T, temporal retina. N, nasal retina.

**Table 1 genes-12-01624-t001:** Demonstrating patients extensive blood work up for nutritional and common genetic causes of optic neuropathy. Blood work up for inflammatory, genetic, and reversible optic neuropathies with normal and patient values.

Lab Test Name	Normal Values	Patient Values
C Reactive Protein (CRP)	<8.0 mg/L	6.2 mg/L
B12	160–950 pg/mL	812 pg/mL
Folate	2.7–17.0 ng/mL	16.8 ng/mL
Methylmalonic acid	87–318 nmol/L	251 nmol/L
Homocysteine	<11.4 μmol/L	11.4 μmol/L
*OPA1*	Associated mutation 2826delT	No mutations
*LHON*	Associated mutations (mt.3460G > A, mt.11778G > A, and mt.14484T > C)	No mutations

## Data Availability

Not applicable.

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
