# Peer review of "WDR36-Associated Neurodegeneration: A Case Report Highlights Possible Mechanisms of Normal Tension Glaucoma"

_genes, 2021, doi:10.3390/genes12101624_

Round 1

Reviewer 1 Report

This paper is a case report of patient with adult-onset primary open angle glaucoma (POAG) with mutations of the WDR36 gene.  Visual function test showed a loss of inner retinal function with preserved photoreceptor function in ERG. The findings recapitulate some of the findings in an animal model of WDR-36 associated POAG.

General comments:

The authors should also show the documentation about the WDR-36 gene defect (not just write about it). They are debating the relationship of different WDR-36 gene mutations in the discussion, so there should be more documentation about the specific gene defect.

Specific comments:

  1. 2, line 50: “in the analyse” – should be “ion the analysis”
  2. 2, line 54-55: “10-year history of … vision loss for about 10 years” – can leave out “for about 10 years” because the 10-year history is mentioned first.
  3. 5, line 125: “the findings partially replicate the findings in an animal model of WDR-associated retinal degeneration” – which animal model?

Figure 1: talks about “few small depigmented round lesions” – could the authors add arrows to the images to point these out?

Figure 2: it is unclear whether this figure has sufficient resolution to show all the details. It would be better if it had page width at 300dpi resolution.

Figure 3: Panels A) and B) may have a too low resolution

Author contributions: lists only 2 authors (A.R. and T.A.). However, the contribution of the first author (E.M.) is not listed.  

Author Response

Thank you very much for your reviews

In response to General review "The authors should also show the documentation about the WDR-36 gene defect (not just write about it). They are debating the relationship of different WDR-36 gene mutations in the discussion, so there should be more documentation about the specific gene defect."

A comprehensive expanded retinal panel that use next generation sequencing to screen for mutations in > 800 genes (panel J905, GeneDx, Gaithersburgh, MD).  This comment has been added to the body of the revised text in lines 72 to 73)

Regarding specific comments

  1. 2, line 50: “in the analyse” – should be “ion the analysis”- this has been changed
  2. 2, line 54-55: “10-year history of … vision loss for about 10 years” – can leave out “for about 10 years” because the 10-year history is mentioned first.- this has been changed 
  3. 5, line 125: “the findings partially replicate the findings in an animal model of WDR-associated retinal degeneration” – which animal model?- A citation has been added to reflect this animal model

Figure 1: talks about “few small depigmented round lesions” – could the authors add arrows to the images to point these out?- blue arrows have been added to point out small subtle round lesions

Figure 2: it is unclear whether this figure has sufficient resolution to show all the details. It would be better if it had page width at 300dpi resolution.- Additional effort was made to create a better resolution.  We are hopeful that the PDF will address this.  It is difficult to remove the actual testing given restrictions of HIPPAA at our hospital.

Figure 3: Panels A) and B) may have a too low resolution- Same issue

Author contributions: lists only 2 authors (A.R. and T.A.). However, the contribution of the first author (E.M.) is not listed.  - E.M.'s contributions have been added.

Reviewer 2 Report

A study by Elana Meer al. (Manuscript ID: Genes-1387027) reports a patient case with phenotype of adult-onset primary open angle glaucoma (POAG) and suggest pathogenic variants in WDR36 gene as a possible cause of normal tension glaucoma in this patient. The pathogenesis of this type of glaucoma is not yet well elucidated and complex mechanism has been suggested by previous studies. The WDR36 gene is suggested to be involved in the process however some conflicting evidence has been reported, which could be related to polymorphism, complexity in pathomechanism, and few cases reported.

This study has few main points:

  1. Presents detailed ophthalmic phenotype of the studied patient using broad range of ophthalmic technologies.
  2. Introduction and case report are generally well presented in the manuscript text.
  3. Novelty of the study is moderate however the importance of publishing this study is high, and may contribute to better understanding of the role of WDR36 gene in the pathogenesis of POAG in the future.

Weaknesses:

  1. Milder phrasing/language edits.
  2. The design of figures is not optimal, would suggest to make few figures out of current Figure 3.

Broad comments:

  1. Please consider changing “mutation” to “pathogenic variant” in the entire manuscript.
  2. Figure 1: enlarged image of the optic discs would be valuable here (if available).
  3. Figure 3: need to be improved (A) colour vision test results do not necessarily need to be shown, results can be described in the text. Kinetic perimetry image need to be improved, this is barely visible. B) were ERGs done according to ISCEV standard protocol? Would suggest that fewer traces are shown. Were mixed rod-cone responses recorded to DA 30.0 or DA 10.0 stimuli? Image with multifocal ERG traces is also barely visible.
  4. The figure legends could include one short sentence about main findings.
  5. Discussion on the pathogenic variant found in the presented patient is missing. A table with pathogenicity scores would be valuable.

Specific comments:

  1. Abstract, row 15: please correct the phrase ‘had a tritan color defect on color vision defect’.
  2. Abstract, row 17: change to ‘associated with’.
  3. Abstract, row 19: change ‘unveiled’ to ‘revealed’.
  4. Row 30: change to ‘worldwide’.
  5. Introduction, row 41: please change to ‘Genetics is considered…’.
  6. Introduction, row 42-43: please change to ‘Many of more previously and extensively studied genes, such as…’
  7. Introduction, row 44-45: make this statement separate one ‘Although, over the past 10 years, it is recognized that these mutations account for less than 10% of patients presenting with disease’.
  8. Introduction, row 50: change to ‘in the analysis’.
  9. Row 67: please rephrase ‘in low teens’.
  10. Row 68: rephrase to ‘Gonioscopic examination revealed open angle in both eyes…’ or similar.
  11. Row 107: change to ‘an explanation’.
  12. Row 110: change to ‘for rod- and cone-specific recordings’.
  13. Row 110-113: please rephrase these two sentences o erg.
  14. Row 113-115: how multifocal ERG confirmed inner retinal dysfunction?? Explain findings from recordings.
  15. Row 115-116: explain which genetic testing was used and which group of genes (panel) was used. What group of genes are those over 800 genes studied?
  16. Row 116: is the variant biallelic homozygous? Correct nomenclature according to the latest recommendations of HGVS, also add transcript number. Please include that this variant was previously published and is classified as pathogenic, reported by gnomAD and ClinVar. Consider changing ‘mutation’ towards ‘pathogenic variant’ in the entire manuscript.
  17. Row 118: remove excessive dots.
  18. Row 156: change to ‘pathogenic variant in WDR36 gene’ and ‘cupping’.
  19. Row 156: change to ‘Eye exam revealed vision 20/20’, remove the first part of the sentence since information about optic disc cupping also appears in this sentence later.
  20. Row 163: write which type of genetic testing was used.
  21. Row 164: change to ‘segregation analysis was not performed due to….’
  22. Row 165: change ‘mutations’ to ‘pathogenic variants’
  23. Row 172-174: please rephrase this sentence.
  24. Row 177: change to ‘Located on the cytogenic band…’.
  25. Row 197-201: change this sentence by writing e.g. There is some contradictory evidence in previously published reports (Ref.)…’ and combine this with the following sentence.
  26. Row 204: write ‘Australian’ , upper-case A.
  27. Row 207: change to ‘abnormalities in WDR36 gene…’
  28. Row 226-227: change to ‘segregation analysis’
  29. Row 228-232: please rephrase this sentence.

Author Response

Thank you for your comments Reviewer 2, particularly as detailed and meticulous as the specific comments.  These specific comments have been addressed and changed within the body of the manuscript.

Regarding the Broad comments, please see our responses that have either been changed, or where we have provided an explanation. 

  1. Please consider changing “mutation” to “pathogenic variant” in the entire manuscript  REPONSE: Yes these have been changed where appropriate in the body of the manuscript.
  2. Figure 1: enlarged image of the optic discs would be valuable here (if available).  REPONSE: Yes, we agree.  The original evaluation was done by photographing the fundus in whole.  We had the patient scheduled to return for dedicated optic nerve evaluation, but he was subsequently lost to follow up, likely due to COVID shut downs.   This is a large reason we provided an OCT image of reliable quality.
  3. Figure 3: need to be improved (A) colour vision test results do not necessarily need to be shown, results can be described in the text. Kinetic perimetry image need to be improved, this is barely visible. B) were ERGs done according to ISCEV standard protocol? Would suggest that fewer traces are shown. Were mixed rod-cone responses recorded to DA 30.0 or DA 10.0 stimuli? Image with multifocal ERG traces is also barely visible.  RESPONSE: 

    We would like to keep the example of the color vision test to illustrate the specific pattern of a tritan defect, which is not what most would expect from a pure optic neuropathy. Please note that the stimuli for each isopter are listed within the legend. The mixed response was elicited with the standard flash, this is now stated in the legend; thank you. A higher resolution image of this figure is now supplied to the journal as a separate file for final insertion by the published (as opposed to embedded within a Word document). Thank you!

  4. The figure legends could include one short sentence about main findings.  RESPONSE: The has been changed in the body of the manuscript
  5. Discussion on the pathogenic variant found in the presented patient is missing. A table with pathogenicity scores would be valuable. RESPONSE: 

    Thank you for pointing this out. We have added a score to the Discussion session. This variant is also predicted to be pathogenic base on a PROVEAN (Protein Variation Effect Analyzer) score of -4.74 (D).

Round 2

Reviewer 1 Report

The authors have edited the paper according to the reviewers’ suggestions and made many improvements. They added more information about the gene defect. However, the reviewer is still not certain whether the figures have sufficient resolution (this could be due to image degradation in pdf).

Figure 1: Even with the arrows, it is impossible to see the small depigmented round lesions. Perhaps it would help to show a normal fundus for comparison.

Figure 2: appears too small, labels are too small (this figure definitively needs to be modified).

Figure 3: partially acceptable (panel C), but some panels have too low resolution.

If the authors could fix the figures, it would be a nice publication.

One small issue in text:

  1. 2 of pdf, paragraph before Table 1: “A comprehensive expanded retinal panel that use next generation sequencing to screen for mutations in > 800 genes (panel J905, GeneDx, Gaithersburgh, MD).” – should be “A … panel used next generation sequencing …”
